# Diagnostic Accuracy of CareStart™ Malaria HRP2 and SD Bioline Pf/PAN for Malaria in Febrile Outpatients in Varying Malaria Transmission Settings in Cameroon

**DOI:** 10.3390/diagnostics11091556

**Published:** 2021-08-27

**Authors:** Innocent Mbulli Ali, Akindeh Mbuh Nji, Jacob Chefor Bonkum, Marcel Nyuylam Moyeh, Guenang Kenfack Carole, Agni Efon, Solange Dabou, Valery Pacome Kom Tchuenkam, Calvino Tah, Jean-Paul Chedjou Kengne, Dorothy Fosah Achu, Jude Daiga Bigoga, Wilfred Fon Mbacham

**Affiliations:** 1The Biotechnology Centre, University of Yaoundé 1, Yaoundé BP 8094, Centre Region, Cameroon; akindeh@gmail.com (A.M.N.); marcel7139@yahoo.com (M.N.M.); tahcalvino@yahoo.com (C.T.); chedjouj@yahoo.fr (J.-P.C.K.); jbigoga@gmail.com (J.D.B.); 2Department of Biochemistry, Faculty of Science, University of Dschang, Dschang BP 67, West Region, Cameroon; kenfackcarole18@yahoo.fr (G.K.C.); alvineagni2@gmail.com (A.E.); dabstars@yahoo.fr (S.D.); pacometchuenkam@gmail.com (V.P.K.T.); 3Department of Animal Biology, Faculty of Science, University of Dschang, Dschang BP 67, West Region, Cameroon; cheforjac@gmail.com; 4Department of Biochemistry, Faculty of Science, University of Buea, Buea BP 63, South West Region, Cameroon; 5National Malaria Control Program, Ministry of Public Health, Yaoundé BP 14386, Centre Region, Cameroon; dollykah@yahoo.com

**Keywords:** malaria, rapid diagnostic test, Histidine-rich protein, sensitivity, CareStart, SD Bioline, Dschang

## Abstract

Background: There was an increase in the number of malaria cases in Cameroon in 2018 that could reflect changes in provider practice, despite effective interventions. In this study, we assessed the diagnostic performance of two malaria rapid diagnostic tests (mRDTs) for diagnostic confirmation of suspected cases of malaria in public and private health facilities in two malaria transmission settings in Cameroon. Methods: We evaluated the diagnostic performance of CareStart pf and SD Bioline Pf/PAN mRDT and compared these parameters by RDT type and transmission setting. Nested PCR and blood film microscopy were used as references. The chi square test was used for independent sample comparisons, while the McNemar’s test was used to test for the dependence of categorical data in paired sample testing. A *p* < 0.05 was considered significant in all comparisons. The R (v.4.0.2) software was used for analyses. Results: A total of 1126 participants consented for the study in the four sites. The diagnostic accuracy of the CareStart Pf mRDT was 0.93.6% (0.911–0.961) in Yaoundé, 0.930% (0.90–0.960) in Ngounso, 0.84% (0.794–0.891) in St Vincent Catholic Hospital Dschang and 0.407 (0.345–0.468) in Dschang district hospital. For SD Bioline Pf/PAN the accuracy was 0.759 (0.738–0.846) for St Vincent Catholic Hospital Dschang and 0.426 (0.372–0.496) for the Dschang district hospital. The accuracy was slightly lower in each case but not statistically different when PCR was considered as the reference. The likelihood ratios of the positive and negative tests were high in the high transmission settings of Yaoundé (10.99 (6.24–19.35)) and Ngounso (14.40 (7.89–26.28)) compared to the low transmission settings of Dschang (0.71 (0.37–1.37)) and St Vincent Catholic hospital (7.37 (4.32−12.59)). There was a high degree of agreement between the tests in Yaoundé (Cohen’s Kappa: 0.85 ± 0.05 (0.7–0.95)) and Ngounso (Cohen’s Kappa: 0.86 ± 0.05 (0.74, 0.97)) and moderate agreement in St Vincent hospital Dschang (k: 0.58 ± 0.06 (0.44–0.71)) and poor agreement in the District Hospital Dschang (Cohen’s Kappa: −0.11 ± 0.05 (−0.21–0.01)). The diagnostic indicators of the SD Bioline Pf/PAN were slightly better than for CareStart Pf mRDT in St Vincent Catholic hospital Dschang, irrespective of the reference test. Conclusions: Publicly procured malaria rapid diagnostic tests in Cameroon have maintained high accuracy (91–94%) in the clinical diagnosis of malaria in high malaria transmission regions of Cameroon, although they failed to reach WHO standards. We observed an exception in the low transmission region of Dschang, West region, where the accuracy tended to be lower and variable between facilities located in this town. These results underscore the importance of the routine monitoring of the quality and performance of malaria RDTs in diverse settings in malaria endemic areas.

## 1. Introduction

Malaria remains a daunting public health challenge worldwide [1]. Despite the fact that significant investments have led to a decline in the mortality and morbidity in malaria in endemic countries, Africa continues to bear the greatest burden. Children less than five years old and pregnant women are impacted more compared to other members of the population. Based on the WHO malaria report of November 2020, there were 229 million cases of malaria infection worldwide, with 409,000 deaths and 94% occurring in Sub-Saharan Africa in 2019. In Cameroon, malaria is also one of the leading causes of mortality and morbidity. Although the risk profile is unevenly distributed across the country, everyone is at risk of malaria parasite infection and disease. The malaria case management policy has evolved to include rapid antigen capture immunochromatographic tests for the parasitological confirmation of malaria among suspected cases in malaria endemic countries (https://www.who.int/teams/global-malaria-programme/case-management/diagnosis, accessed on 2 June 2021) [2]. The use of RDTs offers a simple way of diagnostic testing for malaria in remote areas where capacity for quality assured microscopy is lacking [3]. Other benefits of mRDT are improved reliability in testing compared to microscopy, affordability and quick processing times that enable quick clinical judgement and therapy for a rapidly evolving disease [4]. The malaria RDTs widely used are based on the detection of parasite-specific proteins. The *Plasmodium. falciparum* histidine rich protein II and Plasmodium specific lactate dehydrogenase are the most common of these proteins. Most RDTs in the market are of the lateral flow format and may detect HRP-II only (Pf mRDT) or both HRP II and LDH combined in the same format (Pf/PAN mRDT). The LDH is produced by all common species of the malaria parasite. In the combined RDT format, monoclonal antibodies of the isoforms of each of the common species of the malaria parasite are used on the test strip. Therefore, Pf/PAN mRDTs detect both *P. falciparum* and other species (Pf/PAN). Cameroon adopted the use of RDTs ten years ago, and despite significant gains that have been associated with the implementation of this policy, several challenges have emerged. First, there is a perception among health care providers that mRDTs are not accurate in the diagnosis of malaria among suspected cases. Indeed, one study in Yaoundé, Cameroon noted a low sensitivity of HRP−2 based mRDTs when parasite density is low [5], although in a subsequent study, the difference in sensitivity at a high and a low parasite threshold was not significantly wide [6]. Care providers, thus, continue to treat mRDT negative cases with artemisinin-based combinations as has been observed in other settings [7,8]. Mfuh et al., in studies carried out in three regions in Cameroon between 2013–2015 [9], noted that about 41% of febrile cases treated for malaria were actually parasite negative, which may be related to the entrenched clinical management paradigms, as have been observed in some other malaria endemic countries [8,10,11,12]. One immediate consequence is the observation made by the National Malaria Control Program (unpublished) that in some settings in Cameroon, mRDT and microscopy tests are prescribed for the same suspected case at the same time or in sequence, depending on the outcome of the mRDT test. Secondly, most providers, including private providers, prefer microscopy to mRDTs for several reasons, arguing that implementing mRDT should include a cost recovery component. Thirdly, the supply of mRDTs from the central government warehouse has not been constant, leading to the invasion of the markets by private retailers, increasing the risk of procuring sub-standard mRDT tests by health care facilities. Several factors might affect the performance of rapid diagnostic tests including geography, parasite species distribution, the type of mRDT test, deletions in the *P. falciparum* histidine rich protein antigen in rapid malaria diagnostic tests as well as factors associated with mRDT transportation, storage, and adherence [3,11,13,14] to manufacturer guidelines. The present study was undertaken to evaluate the diagnostic performance of two types of mRDTs (detecting HRP-2 only and HRP-2 and PAN species) used in the public and private sectors in different geographical settings in Cameroon between 2018–2020.

## 2. Methods

### 2.1. Ethical Considerations

The protocol of this study, associated informed consent forms, questionnaires, participant education and recruitment materials, and other study forms, were submitted to the IRB of the Cameroon Baptist Convention Health Services (CBCHS-IRB) with respect to ethical and scientific compliance with applicable research and human subject regulations. Written informed consent was be obtained for each study participant. An explanation of study procedures and implications of study participation (risks/benefits) including potential use of study information for scholarly publication was provided. Consent was obtained after basic eligibility had been established. Participants were provided with adequate time to consider the information provided about the study and to ask additional questions before deciding to participate. For non-mature minors, the study details were explained to at least one parent/guardian for parental permission and then to the minor for assent. For infants, the study details were explained to at least one parent/guardian for parental permission. Mature minors who were able to consent for themselves (students), were allowed to provide self-consent, but they were advised to speak with and involve their parents/guardians or other adults in the study participation process if they felt it was safe to do so. All participant data were stored securely and were only released after permission from study participants. An ethical clearance (reference IRB2019-40) was provided by the CBCHS-IRB before the study started. 

### 2.2. Study Sites

The study sites include four hospitals from two regions in Cameroon. The sites are described and listed below (see map in Appendix A). 

Yaoundé: In Yaoundé, the study took place in the town of Mfou, situated 30 min by car from Yaoundé. This rural setting is located at 4°27′ N and 11°38′ E, in the Centre region of Cameroon. The climate is typically equatorial with two discontinuous dry and wet seasons. The annual average rainfall measures 1727 mm with an average temperature of 23 °C (https://en.climate-data.org/africa/cameroon/centre/yaounde-3987/, accessed on 1 January 2021). 

Ngounso: Located 2 km from the small town of Magba in the Noun division in the West region of Cameroon. It is situated at 5°57′ N and 11°13′ E. The climate is equatorial, characterised by a tropical transition of the Sudano-Guinean type, generally with a long rainy season from mid-March to mid-November and a short dry season from mid-November to mid-March (http://pndp.org/documents/Rapport_CCAP_Magba_final.pdf, accessed on 1 January 2021). Average annual temperature and rainfall are 27 °C and 539 mm, respectively. 

Dschang: Dschang is situated in the Menoua division of the West region of Cameroon. Its geographical coordinates are 5°27′ N and 10°4′ E. It has an equatorial monsoon climate with alternate mountains and valleys determined by the altitude averaging 1400 m (http://www.cvuc-uccc.com/national/index.php/fr/carte-communale/region-de-louest/158-association/carte-administrative/ouest/menoua/458-dschang, accessed on 1 January 2021). It is characterised by a rainy season extending from mid-March to mid-November and a dry season extending from mid-November to mid-March. This town forms part of the major western highlands of Cameroon. 

Malaria transmission is variable in the region, being higher and perennial in the Noun division (Ngounso) compared to Menoua (Dschang). The sites chosen represented the major transmission blocks in the southern part of Cameroon. Additionally, Dschang was selected because it falls within a low transmission transect in the Western Highlands of Cameroon and semi-urban with respect to Yaoundé. Ngounso is a rural locality with seasonal malaria transmission and situated at a lower altitude compared to Dschang.

### 2.3. Study Design

This diagnostic accuracy study was a prospective cross-sectional study in which consenting volunteers suspected of malaria were consecutively enrolled to test for malaria infection using the mRDTs, microscopy and PCR. 

### 2.4. Participants and Sampling

Eligibility: All suspected cases of malaria infection, as defined by the national guideline for malaria case management, from individuals who provided written consent were enrolled into the study. In Cameroon, a suspected case is any individual who reports to the health facility with fever (axillary temperature > 37.5 °C or history of fever with 24 h) and any or more of the following symptoms: lethargic, chills, nausea, joint ache, general body weakness. Participants were identified among all those who consulted clinical staff and were sent to the laboratory for a malaria test. The clinical staff had no role in the study and consent to study procedures was obtained in the laboratory environment by a research assistant. The facilities were both outpatients and inpatients and were recruited in SU-Yaoundé, Dschang and CBCHS-IHC Ngounso between March 2019 and July 2020. This period corresponds to two rainy seasons in the locations and, therefore, periods of moderate to high malaria transmission. The study was interrupted during the dry season between December and February. Participants were sampled consecutively and administered consent and/or assent. Those who accepted were enrolled and then tested for malaria. The sample size for this study was determined using the following formula and considerations. We considered that the Pf test should be at least 90% sensitive in detecting malaria infection among suspected cases. We choose a sample size n, such that the confidence interval is maintained within 5% in an infinite population. By using the following formula:n = Z_α/2_^2^ × p × (1 − p)/d^2^

where n = sample size, Z_α/2_ is the critical value of the normal distribution at α/2 where α = 0.05 for a confidence interval at 95%, p = estimated sensitivity = 90% and d = precision = 5%. We, thus, estimated 138 malaria infected participants. We further adjusted for non-response rate (10%) as well as for those who may not fulfil the inclusion criteria (10%) among eligible patients. This adjusted the sample size to a minimum of 166 malaria positive patients. Given that the prevalence of malaria using microscopy obtained from health facility data was 60% in the same period (April) in 2018, we projected to enrol a minimum of 266 suspected cases of malaria in each of the study areas. 

### 2.5. Test Methods

Index testing: Each participant agreed to provide about 0.5 mL of blood from finger prick, of which 200 µL was used for dried blood spots, 5 µL for each of two types of mRDTs and 20 µL for slide microscopy. The rest was spotted on a Whatman N0 3 filter paper for PCR testing.

Two mRDTs were used for this assessment.
CareStart™ malaria Pf (HRP2) Ag RDT (ACCESSBIO, Somerset, NJ, USA). This mRDT detects parasite histidine rich protein (HRP-2), specifically secreted by *Plasmodium. falciparum*, which is the major parasite species in most malaria infections in Cameroon.SD Bioline Pf/PAN mRDT (Standard DiagnosticsYongin-si, Geonggi-do, Republic of Korea). This is a combination test that detects two proteins viz. HRP-2 and Plasmodium lactate dehydrogenase (pLDH), which is shared by all known Plasmodium species in malaria infections.

Both types of lateral flow immunochromatographic tests rely on the capture of dye labelled antigen–antibody complex by a fixed monoclonal antibody to produce a visible line on a strip of nitrocellulose inside a cassette [15]. The blood sample is drawn by capillary action along the nitrocellulose strip and chased with a buffer solution. The result window on the mRDT is divided into a test window and a control window. The control line that captures excess antibodies bound to dye or antigen–antibody complex gives information on the integrity of the antibody-gold particle conjugate and, therefore, acts as an internal procedural control. Rapid test results without a control line were considered invalid and repeated for the sample in question within 20 min of the invalid test result. 

Both mRDTs are routinely used in health facilities in Cameroon for malaria diagnosis. The Pf mRDT is recommended for use by community health workers to detect malaria infections among suspected cases in the community while the Pf/PAN mRDT is used in health centres and hospitals for parasitological confirmation of malaria among suspected cases. Prior to the deployment of the test system in Cameroon, community health workers as well as laboratory technicians and other health workers at emergency units were trained on the general protocol for the testing and interpreting of mRDTs test outcomes. The retraining of study technicians was performed before study implementation. We compared the diagnostic performance of different tests separately using the standard Giemsa-stained microscopy or PCR as reference (for Yaoundé and Ngounso). This molecular test was additionally chosen because it has documented high sensitivity and specificity in detecting sub-microscopic parasitaemia compared to microscopy, although it is not used routinely. We further reasoned with reducing parasite rates, in the near future, the PCR test may become a useful reference when the National Malaria Control Program re-orientates its strategy towards pre-elimination. Therefore, it was important to set a baseline.

For microscopy, slides were prepared and read by a trained technician in each site. Approximately 6 µL of blood was placed on a clean grease-free slide, about 1 cm from the frosted end and a circle of about 1 cm in diameter of thick film was made after defribrinating the blood through spreading anti-clockwise for about two min. The slide was left to dry in open air away from flies for about one hour. It was later stained with 10% freshly prepared Giemsa for 15 min. The slide was thereafter washed by allowing slow running tap water to wash off the stain with the slide indented about 30 degrees to the running water. The slide was left to dry completely and observed under a binocular microscope in oil immersion. The criteria for stopping the reading was the detection or not of malaria parasites. Slides with low density parasitaemia were read for at least 10 min, whereas slides with high density parasitaemia were read for 2 min or less. This was sufficient to detect malaria parasites on stained slides. In the case of discordant readings, the result of a second study microscopist served as a tie breaker.

For samples that underwent the PCR test, the test was based on amplification of the *P. falciparum* msp-2 gene. Briefly, parasite DNA was extracted using the Chelex-100 method as previously described [16] and amplified through nested PCR using primers and primer conditions specified in Table 1 below.

Each tube consisted of 18.25 µL of nuclease-free water, 2.5 µL of 10× Thermopol buffer, 0.5 µL of dNTPs, 0.25 µL of each external primer (S2 and S3) at a concentration of 2.5 µM, 25 µL of OneTaq^TM^ hot start polymerase at 5 units/µL and 3 µL of DNA extract in a total reaction volume of 25 µL. The amplification conditions were as follows: pre-denaturation at 94 °C for 3 min; denaturation at 94 °C for 30 s, primer annealing at 42 ° C for 1 min, elongation at 65 °C for 2 min × 30 cycles and termination at 72 °C for 3 min. For nested PCR, each tube consisted of 20.25 µL of nuclease-free water, 2.5 µL of 10× Thermopol buffer, 0.5 µL of dNTPs, 0.25 µL of each primer (S1 and S4) at a concentration of 2.5 µM, 0, 25 µL of OneTaq^TM^ hot start polymerase (5 units/µL) and 1 µL of amplicon from the first amplification, in a total volume of 25 µL. The following amplification conditions were applied: denaturation at 94 °C for 30 s, primer annealing at 50 °C for 1 min, elongation at 72 °C for 2 min × 30 cycles and termination at 72 °C for 3 min. 

The amplicons were analyzed using electrophoresis through 2% (wt/vol) agarose gels stained with ethidium bromide. The DNA was visualized under UV.

The expected product size of 1200 bp was measured against a 100 bp DNA ladder (New Englands Biolabs, Ipswich, MA, USA) with a range of fragments from 1000–1517 bp. Samples that had the 1200 bp band were classified as positive and samples that did not have the 1200 band were classified as negative. 

### 2.6. Test Outcome Classification

A positive test was defined as the presence of a red line on the test window in addition to a clearly visible control line. In the absence of a control line, the test was considered invalid and repeated. A negative test was defined as the presence of a red line on the control window only. Thus, for a Pf/PAN test, the result could be either (1) 3 bands on test (Pf line plus pLDH line) and control windows. In this case, the test was interpreted as the presence of *P. falciparum* only or in a mixed infection with other species; (2) two bands on pLDH and control lines. In this case, the test outcome was interpreted as positive for other species than *P. falciparum* but was not used for diagnostic calculations. Since these tests are qualitative tests, and despite the fact that the intensity of the bands might reflect parasite density, no cut-offs were set. A faint line on results window was interpreted as positive, based on manufacturer recommendations. Two trained assessors performed and read the result of each index test. Microscopy was also read by a trained hospital technician who was blinded to the index test. The presence of a trophozoite was noted as a positive test. The presence of a visible band after nested PCR and gel electrophoresis was considered a positive PCR result. Results of the RDT test was delivered to the patient in sites where mRDT was routinely practiced, while microscopy results were delivered to patients in the hospitals that had the technical platform to perform slide microscopy in routine practice. The PCR testing was retrospective and, therefore, the result was not provided to the patient.

### 2.7. Data Analysis

The primary outcome of this study was the diagnostic accuracy and agreement of HRP2 analyte in HRP2 mRDT, the HRP2 and pLDH combo RDTs and microscopy and/or PCR in each site. The performance parameters (sensitivity and specificity) of the RDTs were compared to manufacturer supplied data. The secondary outcomes were a comparison of the performance of the HRP2 analyte in both mRDT test presentations and microscopy and/or PCR within the same hospital in the West, between sites of different malaria transmission intensities within the West region of Cameroon, and between the West and the Centre regions. We defined the diagnostic accuracy using the following modalities in Table 2.
Sensitivity = The proportion of suspected cases with true malaria infection who have a positive mRDT test result and calculated using the following formula: a/a + c.Specificity = The proportion of suspected cases without malaria who have a negative mRDT test result and calculated using the following formula: d/b + dOverall accuracy = The overall accuracy of each mRDT test was defined as the proportion of correct mRDT assessments as a function of all mRDT assessments in the study population and for each type of reference test. It was calculated using the following formula:


Accuracy = (b+ a)/(a + b + c + d).



4.Predictive value of the positive test = This modality defined the probability of a malaria infection among study volunteers with a positive mRDT test result. It was calculated as follows: a/a + b5.Predictive value of the negative test = This was defined as the probability of malaria free participants with a negative mRDT test result, calculated using the following formula: d/c + d.6.The likelihood ratios (LR) of the positive and negative tests were calculated and post-test probabilities expressed as percentages estimated from Bayesian theorem. Thus,



LR+ = sensitivity/(1 − specificity) = (a/(a + c))/(b/(b + d))
LR− = (1 − sensitivity)/specificity = (c/(a + c))/(d/(b + d))
Post-test odds = pre-test odds × LR


## 3. Results

The study enrolled a total population of 1126 participants in the four sites over the study period. In all the sites, there were more females than males marked in the sites in the West region compared to the site in SU-Yaoundé. In CBCHS-IHC Ngounso, West region. However, this difference was not significant. The majority of the study participants were aged 20–40 in the Western regional sites (mean age 31.59 ± 19.6 in Dschang centre, 31.15 ± 17.2 in the St Vincent Catholic hospital catchment area and 26.9 ± 18.4 in Ngounso), followed by those in the age group 21–40. In SU-Yaoundé, individuals 20 years or under formed three quarters of the study participants followed by those 20 years and under (Table 3).
Diagnostic accuracy of CareStart™ malaria HRP2 and SD Bioline Pf/PAN mRDTs in different study sites.Performance of two band mRDT (HRP-2 only) against microscopy as gold standard.We determined the performance characteristics of the two band mRDT rapid tests that detect HRP-2 antigen in patient samples. Table 4 shows the results obtained after analysis.

As shown in Table 4, the Pf RDT sensitivity was high in SU-Yaoundé (0.94; 95% CI (0.91- 0.97)) and CBCHS-IHC Ngounso (0.92; 95% CI (0.86–0.96)), while in the town of Dschang, it was 0.65 (95% CI: 51–77), and low in the St Vincent Catholic Hospital (0.17; 95% CI (0.10–0.26)). The specificity varied following the same trend, although much similar between the different study sites. These results were reflected in the way the likelihood ratios varied across different sites. The probability of obtaining a positive test outcome after Pf mRDT testing was found to be highest in CBCHS-IHC Ngounso (14.40; 95% CI (7.89, 26.28)), followed by SU-Yaoundé (10.99; 95% CI (6.24, 19.35)) and Dschang District hospital (7.37; 95% CI (4.32, 12.59)). By contrast, there was no difference in the probability to obtain negative test outcome after the Pf mRDT among those who are not sick in St Vincent hospital. We also observe that the agreement, as reflected by the kappa statistic, was high in SU-Yaoundé and CBCHS-IHC Ngounso (0.85 ± 0.05 (0.75, 0.95) and 0.86 ± 0.05 (0.74, 0.97), respectively) and negative in the St Vincent Hospital (−0.057 ± 0.06 (0.16, 0.05)). 

When PCR was taken as the reference, almost the same trend in sensitivity, specificity, likelihood ratios and test agreement were observed. However, we note the there was an increase in the likelihood ratios of the positive test outcomes in SU-Yaoundé and CBCHS-IHC Ngounso, and a slight reduction in St Vincent Catholic hospital and the District Hospital Dschang (Table 4). The same trend is observed when considering the diagnostic accuracy in different sites. The accuracy slightly reduced but remained high (>0.90) in SU-Yaoundé, CBCHS-IHC Ngounso, followed by StVC Dschang (0.8) and the district hospital in Dschang (0.34).

Since Pf/PAN mRDT was evaluated in the two sites in the Dschang but not in Yaoundé, the performance of this test in these three sites is reported below.

As shown in Table 5 the Pf/PAN test performed poorly in the DH Dschang, but moderately in StVC Dschang. The sensitivity was 0.23 (95% CI: 0.17–0.31) when compared against microscopy and 0.25 (95% CI: 0.18–0.32) when compared against PCR as referent. Although the predictive values appear to be comparable, the probability of the Pf/PAN mRDT test to be positive among sick participants was appreciably higher in StVC Dschang compared to DH Dschang (4.76(2.84–7.98) versus 0.92 (0.58–1.44)) when compared against microscopy and PCR (4.77 (2.72–8.34) versus 1.07 (0.66–1.73)). The test agreement was also very poor in DH Dschang (−0.02 ± 0.05 (−0.11–0.07) for microscopy) and moderate in StVC Dschang (0.43 ± 0.07 (0.29–0.56) for microscopy) under the same comparison conditions as above. The diagnostic accuracy was higher in the StVC hospital Dschang than in the DH Dschang, irrespective of the gold standard used, but remained less than 0.80.

Given the results, we sought to look at how microscopy performed with PCR as the referent method. The following table presents the results obtained in the four study sites when we compared results of microscopy with those of the PCR test as the reference method.

As shown in Table 6 above, there was comparatively similar trends in sensitivity, specificity, the predictive values of the negative and positive tests as well as in the likelihood ratio of the positive and negative tests. The results indicate that the sensitivity of microscopy, with PCR as the referent, varied from 0.80 (95% CI: 0.69–0.89) in Dschang to 0.92 (95% CI: 0.88–0.95) in SU-Yaoundé. The predictive values of the positive test was high and varied from 0.96 (95% CI: 0.91–0.98) in DH Dschang to 0.99 (95% CI: 0.97–1.00) in SU-Yaoundé. The agreement, as reflected in the kappa statistics, was above in the upper fifth of the index’s classification reflecting perfect agreement between the tests.

We further compared the diagnostic parameters of Pf mRDTs (HRP-2) against Pf/PAN RDTs in each of the two facilities in Dschang when both diagnostic tests were performed on the same participant and on two independent samples, respectively. In each case, we used both microscopy and PCR as gold standards separately. Our analysis showed that Pf/PAN mRDTs performed better in terms of sensitivity and specificity irrespective of the good standard applied. In the same vein, irrespective of the index test used, the test performance was higher in StVC Dschang than in DH Dschang (*p* < 0.05 in each case). 

We observed from Table 6 above that comparatively, Pf/PAN mRDTs performed better in terms of sensitivity and specificity, irrespective of the good standard applied. In the same vein, irrespective of the index test used, the test performance was higher in StVC Dschang than in DH Dschang. 

The performance characteristics of the tests were compared when each of the total number of samples was tested on both mRDTs. In each case, both microscopy and PCR were used as reference standards separately. The results are presented in Table 7 below.

When the Pf mRDT and the combo tests were performed on the same samples in the StVC Dschang (Table 7), no difference in sensitivity was observed, although significant differences in specificity were noted (McNemar’s chi^2^ = 101.14, *p* < 0.001; McNemar’s chi^2^ = 97.3, *p* < 0.001 using microscopy and PCR as reference standards, respectively; see Table 4). By contrast, in the DH Dschang, although both tests had poor sensitivities and specificities, the results were significantly better with the Pf/PAN mRDT compared to the Pf mRDT when microscopy was used as the referent (*p* < 0.001) for sensitivity and specificity comparisons. Likewise, when PCR was used as referent, the same significant differences were obtained (*p* < 0.001 for sensitivity and *p* < 0.02 for specificity).

Furthermore, a pairwise comparison of test parameters taking into consideration all the study sites showed a consistent lower test performance at the DH Dschang, irrespective of the test and referent used (Appendix A).

## 4. Discussion

The goal of this study was to evaluate the diagnostic accuracy and performance characteristics of two malaria rapid diagnostic tests deployed for the diagnosis of malaria between and within health facilities located in semi-urban areas across different malaria transmission settings in Cameroon. The tests were the Pf mRDT (CareStart™ malaria HRP2), which is a two-band test that detects histidine-rich protein 2 secreted by Plasmodium falciparum only, and the three-band test Pf/PAN mRDT (SD Bioline Pf/PAN) that detects both HRP-2 and Plasmodium lactate dehydrogenase (pLDH), the latter secreted by all five main species of malaria parasites infecting humans. All the RDTs were of the lateral flow format and obtained from the same supplier as the study site.

The results show that across different malaria transmission regions, the main malaria RDTs used in Cameroon have remained accurate since they were first deployed in 2011 with few exceptions [6,17]. These findings, however, indicate a slightly lower performance compared to the WHO standards of >95% sensitivity and specificity as well as those reported by the manufacturers. Our results are consistent with findings obtained from similar settings cross malaria endemic Africa [6,9,18,19,20,21].

Our analysis shows that in Dschang, compared to the other study sites, the Pf only and Pf/PAN mRDTs have performed less than expected. In the district hospital of Dschang in particular, the diagnostic parameters were worse. The sensitivity at this site was 17% for the pf only mRDT and 23% for the Pf/PAN mRDT during the study period. The trend could be observed in the St Vincent Catholic hospital where the sensitivity of the pf only mRDT was 65% and 46% for the Pf/PAN mRDT test. Moreover, the predictive values of the positive and negative tests also indicate high rates of positivity and false negative diagnosis. Several reasons could account for this poor quality of malaria RDTs. First, the HRP-2-based RDTs are known to vary with age with decreasing false positivity in older age groups compared to younger age groups [22]. This possibility is ruled out by observing the similarity of the demographic composition of the study participants in the two sites based in the Dschang health district (DH Dschang and StVC Dschang) but that showed significantly different performances based on the same diagnostic parameters. Secondly, the poor performance in DH Dschang could not be related to endemicity or the transmission pattern of the malaria parasite, as was shown in other studies [14,23]. Although Dschang is located in a historically low transmission setting by known definitions, febrile patients who are true cases of malaria will almost invariably be positive upon testing with good quality malaria RDTs due to the absence of substantial clinical immunity. Furthermore, upon inspecting the agreement with the reference tests, we note that the agreement, as reflected in Cohen’s kappa values, was worse in the DH Dschang compared to other sites, excluding a significant influence of sub-microscopic low-density infections that are not picked up by mRDTs. This suggests factors other than differences in transmission patterns should count for the low performance of mRDTs in DH Dschang. Thirdly, the conditions of procurement, transportation and storage of mRDTs in the DH Dschang could affect the diagnostic performance of mRDTs. Several previous reports have associated transportation and storage conditions as key factors affecting mRDT accuracy, such as in Ethiopia, Burkina Faso and Senegal [24]. Gomes et al. (2013) further reported a reduction in the sensitivity and the kappa index for diagnosing falciparum and non-falciparum malaria when RDTs were stored at temperatures other than optimal. Although not formally evaluated, it is likely that this played a role in affecting mRDT performance in the DH Dschang because in the same town a few kilometres away from the DH Dschang, the performance of mRDTs in the StVC Dschang were significantly better, although lower overall than would be expected.

The occurrence of other species than are detected by conventional malaria RDTs in our settings can also affect the performance of such RDTs. In the DH of Dschang, previous reports have indicated the circulation of Plasmodium vivax in the population of febrile patients. An initial [25] and follow up reports within a period of seven years and using molecular tools have demonstrated substantial proportions of Plasmodium vivax in monoinfections and coinfections among malaria cases [26]. Given that this infection often occurs in low densities that may not be readily detected by the PAN pLDH analyte on the conventional Pf/PAN mRDTs deployed in our setting, there remains a likelihood that a substantial amount of malaria infections go unnoticed in this population. It is likely to be missed by microscopy even by well trained technicians in settings where there has historically been no field experience in diagnosing it. Given that there are seasonal variations in the transmission and incidence of malaria, it will be interesting to investigate the occurrence of this species further and assess the value of current tools in the clinical diagnosis of non-falciparum malaria in Dschang.

The ability of the mRDTs to test negative among non-diseased participants was equally appreciably higher in other sites than in Dschang. This is reflected in the predictive values of the positive and negative test results that were consistently poorer. The positive predictive value varies by season and age group and was shown to be higher in children than in adults in Burkina Faso during the malaria transmission season [27]. Although we did not discriminate performance parameters by age groups, our results in Yaoundé lend support to those obtained in the study in Burkina Faso. The highest negative predictive value of the Pf only RDT compared to the other sites was obtained in Ngounso, although the difference did not appear to be statistically significant. This difference may, however, reflect the relative abundance of Plasmodium falciparum parasites in these regions or the presence of the well-known deletion in the HRP-2/HRP-3 gene that affects the detection of the antigen in clinical isolates [28,29].

The likelihood ratio is a useful tool for assessing the effectiveness of a diagnostic test because they determine how much more probable it is to find a positive result in an infected person compared to an uninfected person. Our results further show that in regions of moderate to high transmission of malaria in Cameroon, both mRDTs have a strong ability to predict malaria infections among febrile cases suspected for malaria, as indicated by estimates of post-test probability of disease. The likelihood ratios of the positive test for both the pf only and the Pf/PAN mRDT were high in Ngounso and, Yaoundé and moderate in StVC Dschang. This translates to a post-test probability of >97% for Ngounso and Yaoundé and 91.3% for StVC Dschang. Previous studies have shown that when parasite densities are high, there are greater odds of obtaining a positive test outcome among febrile malaria suspects [23,30]. The decline in these odds among the sites could relate to the parasite density (lower in the Dschang health district) and, hence, the transmission pattern, mRDT procurement practices, and occurrence of other clinical conditions not assessed in the present study such as anaemia or rheumatoid arthritis.

The pattern of mRDT performance across different sites mirrors the altitude patterns compatible with malaria transmission, irrespective of the study population. In regions at a low altitude, such as Ngounso and Yaoundé (about 700 m above sea level), malaria transmission is shown to be high, and RDT performance is appreciably high, although it failed to satisfy the 95% threshold recommended by the WHO. In Dschang (about 1380 m above sea level), on the other hand, the mRDT performance was poorer, although this varied between facilities in the same town. The McNemar’s analysis of paired samples in our population in these two health facilities yielded similar results (with little differences) to the independent sample test, further lending validity to the consistency of the test outcomes after performance and the overall results.

Our study had some limitations, among which we did not conduct a density-dependent analysis of mRDT performance across the sites. However, while this was an interesting option, the limitations in funding did not allow full expert quantification of malaria parasites for over a thousand patients within the timeframe. This is the first study conducted in Cameroon that compares the clinical performance across geographies to provide useful clinical data on RDT performance and guidance on a consideration of regional specificities in some cases in the interpretation of mRDT results. Our results are also useful in that it will help the national malaria control program in its recent quest to implement the national quality assurance program for an accurate and a timely malaria diagnosis in Cameroon. We intend to examine the molecular species contributing to malaria infections among samples from Dschang to further clarify the utility of RDTs in this region. Lastly, it would have been more interesting to use the genus primers for the PCR analysis of malaria positive samples, which we intend to undertake with new funding. Overall, our results may prove is useful to move the program to its ambitious goals of malaria pre-elimination within the current decade.

## 5. Conclusions

The present diagnostic study shows that malaria rapid diagnostic tests in Cameroon have maintained a high accuracy (91–94%) in the clinical diagnosis of malaria across regions of different malaria transmission, although they failed to reach the WHO threshold. These results underscore the importance of routine monitoring of the quality and performance of malaria RDTs in diverse settings in malaria endemic areas.

## Figures and Tables

**Table 1 diagnostics-11-01556-t001:** PCR primers for *pfMSP-2* detection.

Gene and Type de PCR	Primers	Sequence (5′→3′)
Primary PCR primers	S2	GAG GGA TGT TGC TGC TCC ACA G
	S3	GAA GGT AAT TAA AAC ATT GTC
Nested PCR Primers	S1	GAG TAT AAG GAG AAG TAT G
	S4	CTA GAA CCA TGC ATA TGT CC

**Table 2 diagnostics-11-01556-t002:** A 2 × 2 table used to compute indicators of diagnostic accuracy of the two rapid diagnostic tests.

	Reference Test Outcome			
		Positive	Negative	Total
Index test outcome	Positive	a	b	a + b
	Negative	c	d	c + d

Where a = TP: true positive, b = FP: false positive, c = FN: false negative and d = TN: true negative obtained by comparing the mRDT result with the reference standard microscopy or PCR.

**Table 3 diagnostics-11-01556-t003:** Study participant demographic distribution.

Study Sites Parameters		StVC Dschang (217)	DH Dschang (243)	CS CBCHS-IHC Ngounso (286)	SU-Yaoundé (380)
Mean age ± SD		31.59 ± 19.6	31.15 ± 17.2	26.9 ± 18.4	19.4 ± 17.3
% (n)					
Sex	F	76.5 (166)	80.9 (196)	65.4 (187)	56.9 (214)
	M	23.5 (51)	19.1 (46)	34.6 (99)	43.1 (162)
Age groups	(0–20)	23.7 (50)	23.7 (57)	39.5 (113)	67.4 (256)
	(20–40)	50.7 (107)	55.6 (134)	44.4 (127)	18.4 (70)
	(40–60)	15.6 (33)	14.9 (36)	9.8 (28)	11.3 (43)
	(60–80)	8.1 (17)	5.8 (14)	5.2 (15)	2.9 (11)
	>80	1.9 (4)	/	1.0 (3)	/

SD: Standard deviation, n = total population size for each modality.

**Table 4 diagnostics-11-01556-t004:** Performance parameters of Pf mRDT against microscopy and PCR as gold standard in the four study sites.

Study Sites Parameters	Reference Test	StVC Dschang (217)	DH Dschang (243)	CS CBCHS-IHC Ngounso (286)	SU-Yaoundé (380)
Sensitivity	Microscopy	0.65 (51–77)	0.17 (0.10, 0.26)	0.92 (0.86, 0.96)	0.94 (0.91, 0.97)
	PCR	0.56 (0.43, 0.68)	0.14 (0.08, 0.23)	0.86 (0.80, 0.91)	0.91 (0.87, 0.94)
Specificity	Microscopy	0.91 (0.86, 0.95)	0.77 (0.64, 0.87)	0.94 (0.89, 0.97)	0.91 (0.85, 0.96)
	PCR	0.92 (0.86, 0.96)	0.71 (0.57, 0.83)	0.96 (0.92, 0.99)	0.97 (0.92, 0.99)
PPV	Microscopy	0.73 (0.58, 0.84)	0.52 (0.33, 0.71)	0.92 (0.86, 0.96)	0.96 (0.92, 0.98)
	PCR	0.76 (0.63, 0.87)	0.48 (0.29, 0.67)	0.96 (0.91, 0.99)	0.99 (0.97, 1.00)
NPV	Microscopy	0.88 (0.82, 0.92)	0.38 (0.29, 0.47)	0.94 (0.89, 0.97)	0.89 (0.83, 0.94)
	PCR	0.81 (0.74, 0.87)	0.31 (0.23, 0.40)	0.87 (0.81, 0.92)	0.81 (0.73, 0.87)
LR+	Microscopy	7.37 (4.32, 12.59)	0.71 (0.37, 1.37)	14.40 (7.89, 26.28)	10.99 (6.24, 19.35)
	PCR	6.78 (3.79, 12.12)	0.50 (0.26, 0.94)	24.31 (10.25, 57.64)	32.98 (10.80, 100.74)
LR-	Microscopy	0.38 (0.27, 0.55)	1.09 (0.92, 1.29)	0.08 (0.05, 0.15)	0.06 (0.04, 0.10)
	PCR	0.48 (0.37, 0.63)	1.20 (1.00, 1.46)	0.14 (0.10, 0.22)	0.09 (0.07, 0.14)
k ± SE	Microscopy	0.58 ± 0.06 (0.44, 0.71)	−0.057 ± 0.06 (0.16, 0.05)	0.86 ± 0.05 (0.74, 0.97)	0.85 ± 0.05 (0.75, 0.95)
	PCR	0.51 ± 0.07 (0.38, 0.64)	−0.11 ± 0.05 (−0.21, −0.01)	0.82 ± 0.6 (0.71, 0.94)	0.83 ± 0.05 (0.76, 0.93)
Diagnostic accuracy	Microscopy	0.84	0.41	0.93	0.94
	PCR	0.80	0.34	0.91	0.92

PPV: predictive value of positive test; NPV: predictive value of negative test; LR: Likelihood ratio; K: Cohen’s kappa statistic; SE: Standard error.

**Table 5 diagnostics-11-01556-t005:** Performance parameters of Pf/PAN RDT against microscopy and PCR as reference tests.

Study Sites Parameters	StVC Dschang (217)		DH Dschang (243)	
	Reference Test			
	Microscopy	PCR	Microscopy	PCR
Sensitivity	0.51 (0.37, 0.64)	0.46 (0.34, 0.58)	0.23 (0.17, 0.31)	0.25 (0.18, 0.32)
Specificity	0.89 (0.83, 0.94)	0.90 (0.84, 0.95)	0.75 (0.65, 0.83)	0.77 (0.67, 0.86)
PPV	0.63 (0.48, 0.77)	0.70 (0.54, 0.82)	0.59 (0.45, 0.71)	0.67 (0.54, 0.79)
NPV	0.84 (0.77, 0.89)	0.78 (0.71, 0.84)	0.39 (0.32, 0.46)	0.35 (0.28, 0.42)
LR+	4.76 (2.84, 7.98)	4.77 (2.72, 8.34)	0.92 (0.58, 1.44)	1.07 (0.66, 1.73)
LR-	0.55 (0.42, 0.72)	0.60 (0.48, 0.75)	1.03 (0.89, 1.19)	0.98 (0.85, 1.13)
k ± SE	0.43 ± 0.07 (0.29, 0.56)	0.40 ± 0.6 (0.26, 0.52)	−0.02 ± 0.05 (−0.11, 0.07)	0.01 ± 0.4 (−0.07, 0.1)
Diagnostic Accuracy	0.79	0.76	0.43	0.42

PPV: predictive value of positive test; NPV: predictive value of negative test; LR: Likelihood ratio; K: Cohen’s kappa statistic; SE: Standard error.

**Table 6 diagnostics-11-01556-t006:** Performance parameters of microscopy against PCR (Gold standard).

Study Sites Parameters	StVC Dschang (217)	CS CBCHS-IHC Ngounso (286)	SU-Yaoundé (380)
Sensitivity	0.80 (0.69, 0.89)	0.88 (0.81, 0.92)	0.92 (0.88, 0.95)
Specificity	0.99 (0.96, 1.00)	0.98 (0.94, 1.00)	0.98 (0.94, 1.00)
PPV	0.98 (0.91, 1.00)	0.98 (0.93, 1.00)	0.99 (0.97, 1.00)
NPV	0.91 (0.86, 0.95)	0.88 (0.82, 0.93)	0.84 (0.76, 0.90)
LR+	116.00 (16.39, 820.81)	41.17 (13.42, 126.31)	50.28 (12.73, 198.56)
LR-	0.20 (0.13, 0.32)	0.13 (0.08, 0.20)	0.08 (0.05, 0.12)
k ± SE	0.83 ± 0.6 (0.70, 0.96)	0.85 ± 0.59 (0.74, 0.97)	0.86 ± 0.05 (0.76, 0.96)

PPV: predictive value of positive test; NPV: predictive value of negative test; LR: Likelihood ratio; K: Cohen’s kappa statistic; SE: Standard error.

**Table 7 diagnostics-11-01556-t007:** Performance of paired testing against microscopy and PCR in the StVC Dschang and DH Dschang.

Site		StVC Dschang			
GS	Parameters	Index Tests		McNemar’s chi^2^	*p*
		Pf RDT	Pf + PAN RDT		
Microscopy	Sensitivity	0.65 (51–77)	0.56 (0.43, 0.68)	1.65	0.19
	Specificity	0.91 (0.86, 0.95)	0.92 (0.86, 0.96)	101.14	˂0.001
PCR	Sensitivity	0.56 (0.43, 0.68)	0.51 (0.37, 0.64)	0.01	0.8
	Specificity	0.92 (0.86, 0.96)	0.89 (0.83, 0.94)	97.30	˂0.001
**Site**		**DH Dschang**			
**GS**	**Parameters**	**Index Tests**		**McNemar’s chi^2^**	** *p* **
		**Pf mRDT**	**Pf + PAN mRDT**		
Microscopy	Sensitivity	0.17 (0.10, 0.26)	0.23 (0.17, 0.31)	15.42	˂0.001
	Specificity	0.77 (0.64, 0.87)	0.75 (0.65, 0.83)	97.30	˂0.001
PCR	Sensitivity	0.14 (0.08, 0.23)	0.25 (0.18, 0.32)	16.47	˂0.001
	Specificity	0.71 (0.57, 0.83)	0.77 (0.67, 0.86)	4.74	0.02

## Data Availability

Data are contained within the article or Appendix A.

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
