# Peer review of "Diagnostic Accuracy of CareStartâ„¢ Malaria HRP2 and SD Bioline Pf/PAN for Malaria in Febrile Outpatients in Varying Malaria Transmission Settings in Cameroon"

_diagnostics, 2021, doi:10.3390/diagnostics11091556_

Round 1

Reviewer 1 Report

The present manuscript is of potential interest and relevant, but in my view it requires substantial improvements.  The overall manuscript presentation highlight deficiencies that would need to be addressed.

  1. The tables and figure in the manuscript are not correctly identified. Additionally, tables must be mentioned in the text, as well as all tables must have a title.
  2. Descriptions of study sites (line102-120) can be added to Figure 1, making it more informative. If possible use a heat scale to show the epidemiology of malaria in these regions.
  3. The introduction must be improved. Previous data from evaluations already carried out in the country are lacking to evidence the state of the art.
  4. It is not clear why these regions were chosen.
  5. The authors mentioned that: “…Each participant agreed to provide about 0.5ml of blood from finger stick which was divided as follows: 200 ul for dried blood spots, 5ul for each of two types of mRDTs and 20ul for slide microscopy…”. However, in the description of the microscopic analysis the authors detail: “…Approximately 6ul of blood was placed on a clean grease free slide, about 1cm from the…”. Review these sentences.
  6. (Line 206-207) Slides with low density parasitaemia were read for at least ten minutes whereas slides with high density parasitaemia were read sometimes for two minutes or less. More accuracy is needed when expressing these concepts, especially considering previous reports that have established relationships between the parasitemia measurement and diagnosis. Why do authors disregard this? Could false negatives of rapid tests be related to low level parasitemia? Did any false negatives have a high parasite load?
  7. (Line 325-326): …this difference did not seem to be quite marked…Authors should be more precise considering the statistical analyzes performed.
  8. Did the authors collect data from the clinical assessment of the volunteers? Is this information not relevant? 
  9. The presentation of the results is confusing and the disorganization of the tables makes its analysis difficult. A participants’ recruitment flow chart should facilitate understanding of the methodology and results.
  10. Names should always be italicized. Please review the manuscript.

Author Response

The Authors thank the Reviewer 1 for the very helpful comments that will greatly improve the quality of our manuscript. We have attached the detailed response and a revised manuscript.

Sincerely,

Dr Ali, on behalf of co-authors.

Reviewer 2 Report

This paper investigates the effectiveness of the use of rapid immunochromatographic tests for the diagnosis of malaria in Cameroon and presents a large-scale statistical study.
Before this manuscript is published, it needs substantial revision:
1. It is not recommended to use not generally accepted abbreviations (like RDT or mRDT) in the abstract. Moreover, such abbreviations should not be among the keywords.
2. All abbreviations must be explained at the first mention, and not in the middle of the text (see line 94).
3. All figures, tables and formulas should be numbered sequentially.
4. Should there be d instead of b in the formula on line 279? If so, is it a typo in the formula, or is the formula calculated incorrectly as well?
5. Correct the brackets in Table 2.
6. In Tables 2-1, 3-4а and in Supplementary tables, the sensitivity values ​​differ by several orders of magnitude. Check it out. All values ​​should be given in the same dimensions.

Author Response

Dear Reviewer,

The Authors wish to thank you for the time taken to review this manuscript. We have provided a point by point response to all the remarks made and the revisions incorporated in the revised copy of the manuscript. Please consult this version of the manuscript as well. Attached our responses.

SIncerely,

Dr Ali, on behalf of co-authors.

Reviewer 3 Report

Dear Authors,

Please find my observations on your manuscript in attached word document. 

Author Response

Dear Reviewer,

The Authors wish to thank you for the time taken to meticulously review this manuscript. We have provided a point by point response to all the remarks made and the revisions incorporated in the revised copy of the manuscript. Please consult this version of the manuscript as well. Attached our responses.

Sincerely,

Dr Ali, on behalf of all co-authors.

Round 2

Reviewer 1 Report

The manuscript is substantially improved. The authors have addressed most comments, and now I have only a few minor additional suggestions as per below.

  1. Review the identification and order of the tables.
  2. The purpose of figure 1 is unclear. The figure is not informative. It is suggested to add additional information in the figure to better understand the characteristics of the studied sites. 
  3. Please review the values and units presented in the abstract and tables. 

Author Response

We thank the Reviewer for the further comments. Below is a point-by-point response to the comments.

  1. Review the identification and order of the tables.

Authors: We have revised the manuscript to consider this.

  1. The purpose of figure 1 is unclear. The figure is not informative. It is suggested to add additional information in the figure to better understand the characteristics of the studied sites. 

Authors: We have included a caption with additional information. Furthermore, a description follows this map that provides detailed information about the study sites.

  1. Please review the values and units presented in the abstract and tables. 

Authors: We have completely reviewed and corrected the units in both abstract and the tables. Please see revised manuscript in track changes.

Reviewer 2 Report

The manuscript has been greatly improved, but some of my comments have gone unheeded. For example, the values in the tables, both in the main text and in the Supplement, have not been corrected. The authors must decide in what dimensions they give the values of sensitivity and specificity - in %, or in fractions. So in the table on page 13 the sensitivity of microscopy is 65 (51-77), in the table on page 17 the same parameter is 0.65 (51-77). The tables are not numbered correctly. The table on p. 12 the inverted brackets.

Author Response

We thank the Reviewer for the further comments. Our response is presented below.

Reviewer:

The manuscript has been greatly improved, but some of my comments have gone unheeded. For example, the values in the tables, both in the main text and in the Supplement, have not been corrected. The authors must decide in what dimensions they give the values of sensitivity and specificity - in %, or in fractions. So in the table on page 13 the sensitivity of microscopy is 65 (51-77), in the table on page 17 the same parameter is 0.65 (51-77). The tables are not numbered correctly. The table on p. 12 the inverted brackets.

Authors: 

We have revised the manuscript and corrected the units and made this consistent through the abstract, manuscript and tables. 

Reviewer 3 Report

Dear Authors,

I am sure you have effected corrections on the concerns raised during review round 1, but I cannot independently verify such. I can only do it if you provide a tracked changes version of the manuscript showing the changes you have effected. Also, in your rebuttal letter where changes have been done to indicate the line number(s), please. 

Author Response

We thank the Reviewer for this comment. Our response as below.

Reviewer:

I am sure you have effected corrections on the concerns raised during review round 1, but I cannot independently verify such. I can only do it if you provide a tracked changes version of the manuscript showing the changes you have effected. Also, in your rebuttal letter where changes have been done to indicate the line number(s), please. 

Authors.

Please find the manuscript uploaded here in track changes, and in the main portal in this system the clean copy. We will be glad to have your further reviews, if any.
